# Production location of the gelling agent Phytagel has a significant impact on *Arabidopsis thaliana* seedling phenotypic analysis

Caitlin N. Jacques[1,2], Anna K. Hulbert[2☯], Shelby Westenskow[2☯], Michael M. Neff [1,2]*

**1** Graduate Program in Molecular Plant Sciences, Washington State University, Pullman, WA, United States of America, **2** The Department of Crop and Soil Sciences, Washington State University, Pullman, WA, United States of America

☯ These authors contributed equally to this work.
* mmneff@wsu.edu

**Data Availability Statement:** All relevant data are within the manuscript and its Supporting Information files.

## Abstract

### Background

Recently, it was found that 1% Phytagel plates used to conduct *Arabidopsis thaliana* seedling phenotypic analysis no longer reproduced previously published results. This Phytagel, which is produced in China (Phytagel C), has replace American-made Phytagel (Phytagel), which is no longer commercially available. In this study, we present the impact of Phytagel produced in the United States vs. China on seedling phenotypic analysis. As a part of this study, an alternative gelling agent has been identified that is capable of reproducing previously published seedling morphometrics.

### Results

Phytagel and Phytagel C were investigated based on their ability to reproduce the subtle phenotype of the *sob3-4 esc-8* double mutant. Fluence-rate-response analysis of seedlings grown on 1% Phytagel C plates failed to replicate the *sob3-4 esc-8* subtle phenotype seen on 1% Phytagel. Furthermore, root penetrance analysis showed a significant difference between *sob3-4 esc-8* seedlings grown on 1% Phytagel and 1% Phytagel C. It was also found that 1% Phytagel C was significantly harder than 1% Phytagel. As a replacement for Phytagel C, Gellan was tested. 1% Gellan was able to reproduce the subtle phenotype of *sob3-4 esc-8*. Furthermore, there was no significant difference in root penetration of the wild type or *sob3-4 esc-8* seedlings between 1% Phytagel and 1% Gellan. This may be due to the significant reduction in hardness in 1% Gellan plates compared to 1% Phytagel plates. Finally, we tested additional concentrations of Gellan and found that seedlings on 0.6% Gellan looked more uniform while also being able to reproduce previously published results.

**Funding:** This work was supported by the United States National Science Foundation Project #1656265 (to M.M.N.). This work was also supported by the USDA National Institute of Food and Agriculture, Hatch Umbrella Project #1015621 (to M.M.N.).

## Conclusions

Phytagel has been the standard gelling agent for several studies involving the characterization of subtle seedling phenotypes. After production was moved to China, Phytagel C was no longer capable of reproducing these previously published results. An alternative gelling agent, Gellan, was able to reproduce previously published seedling phenotypes at both 1% and 0.6% concentrations. The information provided in this manuscript is beneficial to the scientific community as whole, specifically phenomics labs, as it details key problematic differences between gelling agents that should be performing identically (Phytagel and Phytagel C).

## Background

The study of phenomics in *Arabidopsis thaliana* (*A. thaliana*) is the focus of many molecular and physiology labs worldwide. One of the ways that *A. thaliana* growth and development can be studied is through the use of growth media plates. The use of growth media plates for the study of *A. thaliana* has many benefits, including affordability, transparency, ease, and most importantly, reproducibility.

Growth media plates are made with agar derived from red algae, or more commonly, by agar substitutes. One common agar substitutes is Phytagel (Sigma). Phytagel is produced from a bacterial substrate that is composed of rhamnose, glucuronic acid, and glucose [1]. Phytagel creates a clear, colorless growth matrix for plants. Another widely used agar substitute is gellan gum. Gellan Gum (Gellan) (PhytoTechnology Laboratories, Inc.) is produced by bacterial fermentation of *Sphingomonas elodea*, which creates a high molecular weight polysaccharide gum. This gum is composed of repeating tetrasaccharide units that will form a gel in the presence of mono- or divalent cations [2].

Growth media plates allow for the seeds of small plants, such as *A. thaliana*, to be grown and phenotyped in controlled environments. These plates are especially important when studying *A. thaliana* plants with subtle phenotypes. An activation tagging screen was conducted to identify suppressors of the long hypocotyl phenotype conferred by the weak, missense *phyB-4* mutant allele [3–9]. From the activation tagging screen, *SUPPRESSOR OF PHYTOCHROME B-4 #3* (*SOB3*) and its closest paralog, *ESCAROLA* (*ESC*), were identified [3–4, 9]. Null alleles of both *SOB3* and *ESC* were identified as *sob3-4* and *esc-8*, respectively [9]. It was observed that the *sob3-4 esc-8* double mutant produced a subtle hypocotyl phenotype that was taller than the wild type (WT), but shorter than the extreme-tall phenotype of the dominant-negative *sob3-6* allele [9,10].

1% Phytagel plates containing 1.5% sucrose were used as a standard to conduct all of the aforementioned phenomics research. Additionally, 1% Phytagel plates have been used to reproduce the results of the *sob3-6* mutant [11], as well as in new research with the subtle phenotypes of *A. thaliana* NAC Domain Containing Protein 81 (ATAF2) mutants [12]. Therefore, 1% Phytagel plates are sufficient for phenotypic analysis of subtle mutant phenotypes and the results are reproducible.

In 2016, we discovered issues with our standard growth media plates. The plates were harder to the touch, the seedlings appeared to be germinating asynchronously, and did not look healthy. After notifying Sigma of these issues and receiving Phytagel from a different lot number, the problems persisted. We soon discovered that both of the new batches of Phytagel

were now being produced in China (Phytagel C). We compared growth of *A. thaliana* seedlings on plates made with American-made Phytagel (Phytagel) to their growth on plates made with Phytagel C. We included another gelling agent, Gellan, produced by PhytoTechnology Laboratories Inc., in our experimentation. The purpose of these experiments was to explore how different gelling agents performed under light intensities commonly used for seedlings phenomics, and to possibly indicate the gelling agent that produced the most uniform germination, root penetrance, and/or hypocotyl fluence rate responses. In this study we present the impact of Phytagel produced in the United States vs. China on seedling phenotypic analysis. As a part of this study, an alternative gelling agent has been identified that is capable of reproducing previously published seedling morphometrics.

## Results

Hypocotyl length measurements, as well as fluence-rate-response analysis, have been used to elucidate the subtle mutant phenotypes, such as *phyB-4* [3] and *sob3-4 esc-8* [9]. The standard for these experiments is 1% Phytagel media [3,9,11]. The differences between these subtle phenotypes can be best observed at a white light intensity of 10 µmol m$^{-2}$s$^{-1}$ (Fig 1; Fig 2, 2A and 2C) [9]. The *sob3-4 esc-8* double mutant has been observed to be significantly taller than the WT, but shorter than *sob3-6* [9]. This difference in average hypocotyl length between WT and *sob3-4 esc-8* on 1% American-made Phytagel is statistically significant (Fig 2 and 2B), but this difference is not statistically significant on 1% Phytagel C media (Fig 2 and 2D). However, both Phytagel and Phytagel C are able to distinguish the difference between the WT and more severe mutant phenotypes conferred by *SOB3-D* and *sob3-6* (Fig 2, 2B and 2D). Interestingly, WT dark grown seedlings on Phytagel and Phytagel C media are significantly different, but no such significance is observed for dark grown *sob3-4 esc-8* seedlings on Phytagel and Phytagel C media (see S2 Fig). In addition, the length of the hypocotyls for all genotypes are significantly different (at least $p < 0.01$) between 1% Phytagel C media and 1% Phytagel media (Fig 2, 2B and 2D). This led us to suspect that there may be an issue with seedling germination or seedling development on Phytagel C media. Therefore, we conducted germination and root penetrance assays to determine if either of these factors are impacted by Phytagel C media.

Germination rates were calculated for each of the four genotypes (WT, *sob3-6*, *SOB3-D*, and *sob3-4 esc-8*) on 1% Phytagel and 1% Phytagel C plates at different light intensities (10, 60, and 100 µmol m$^{-2}$ s$^{-1}$). No trends were observed that would indicate a clear connection between gelling agent and percent germination (see S1 Fig). However, root penetrance was impacted by growth on 1% Phytagel C media. It was found that there is no significant difference between Phytagel and Phytagel C at 10 and 100 µmol m$^{-2}$s$^{-1}$ for the WT, but there is a significant difference at 60 µmol m$^{-2}$s$^{-1}$ (Fig 3 and 3A). Furthermore, the average percent root penetrance for *sob3-4 esc-8* at all light intensities were statistically different between 1% Phytagel and 1% Phytagel C plates (Fig 3 and 3B), suggesting that there is physical difference between these two gelling agents. In order to test if the Phytagel was physically different from Phytagel C, a hardness analysis was performed (Fig 4). The 1% Phytagel C plates were significantly harder than the 1% Phytagel plates (Fig 4), which may explain the root penetrance data in Fig 3.

Since the original Phytagel is no longer available for purchase, and Phytagel C has an adverse impact on seedling phenomics, another gelling agent, Gellan, was compared to Phytagel. Gellan was able to distinguish the subtle phenotype of *sob3-4 esc-8* through fluence-rate-response analysis (Fig 5, 5A and 5C) and was able to significantly separate *sob3-4 esc-8* from the WT (Fig 5, 5B and 5D).

Percent root penetrance on 1% Gellan plates were not negatively impacted for any of the genotypes at any light fluence-rate when compared to 1% Phytagel plates (Fig 6 and 6A). In

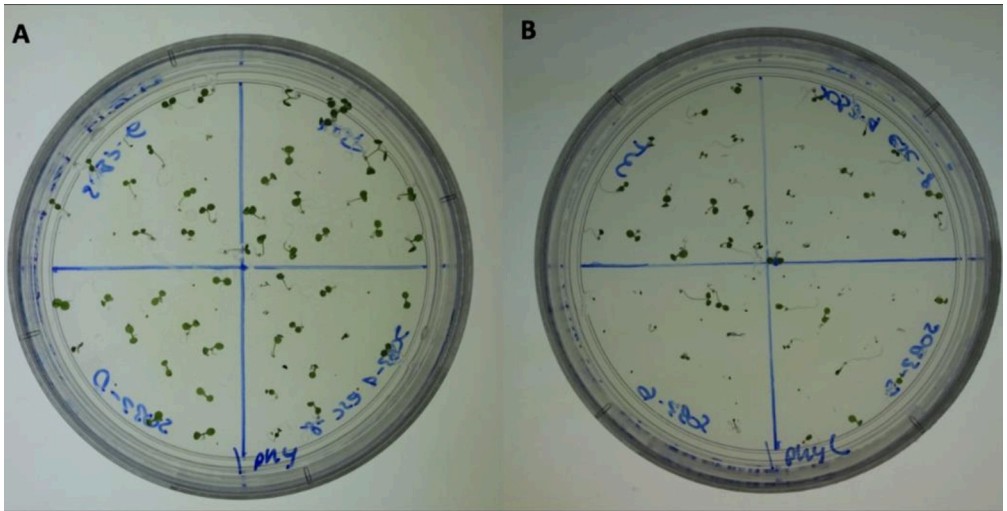

**Fig 1. Pictures of seedlings grown on 1% media at 10 µmol m$^{-2}$s$^{-1}$ for six days. A)** 6-day-old seedlings grown on 1% Phytagel at 10 µmol m$^{-2}$s$^{-1}$. **B)** 6-day-old seedlings grown on 1% Phytagel C at 10 µmol m$^{-2}$s$^{-1}$.

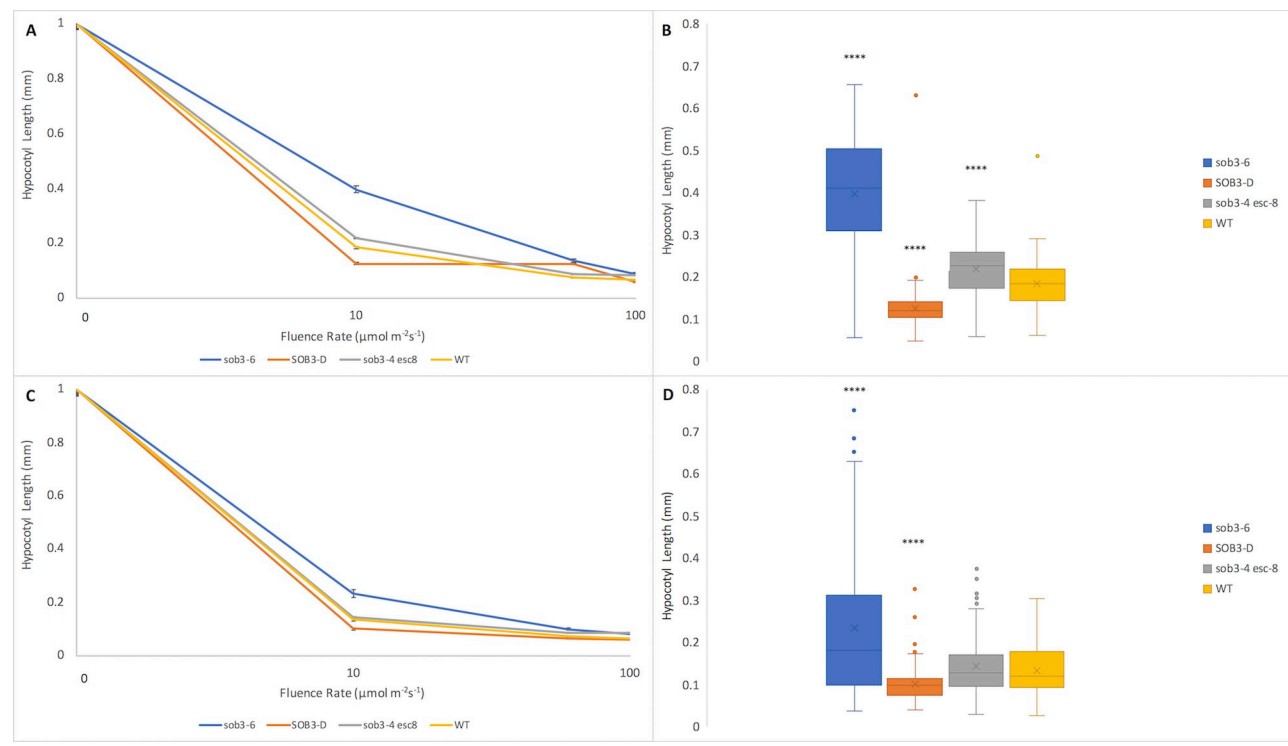

**Fig 2. Fluence rate responses of 6-day-old seedlings on 1% Phytagel and 1% Phytagel C plates. A)** Fluence rate responses of 6-day-old seedlings that have been normalized to the dark control on plates containing 1% Phytagel media. **B)** Graphical representation of 6-day-old seedlings that have been normalized to the dark control on plates containing 1% Phytagel media at 10 µmol m$^{-2}$s$^{-1}$. **C)** Fluence rate responses of 6-day-old seedlings that have been normalized to the dark control on plates containing 1% Phytagel C media. **D)** Graphical representation of 6-day-old seedlings that have been normalized to the dark control on plates containing 1% Phytagel C media at 10 µmol m$^{-2}$s$^{-1}$. Standard error is shown for all data sets. In a Welch's t test (unpaired two-tailed t test with unequal variance) compared with the wild type: P > 0.05 = NS, P ≤ 0.0001 = ****.

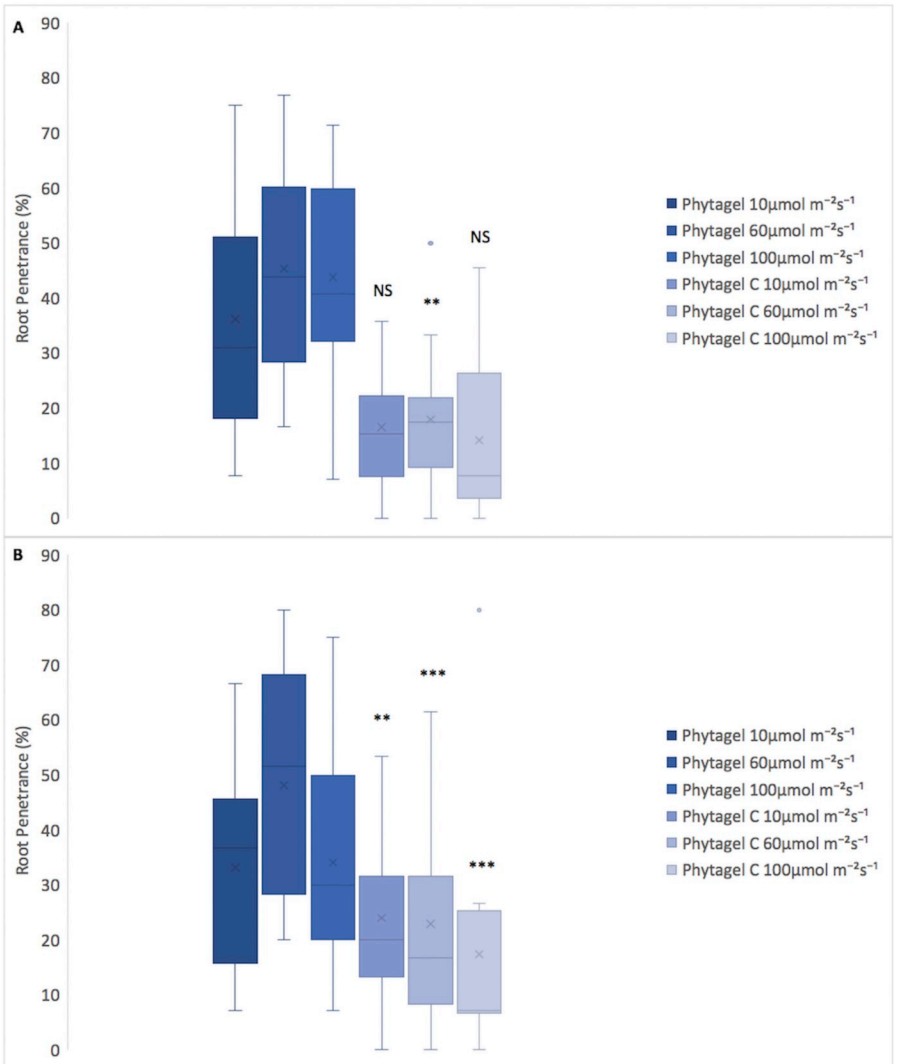

**Fig 3. Average root penetrance of 6-day-old seedlings on 1% Phytagel and 1% Phytagel C plates. A)** Average root penetrance of 6-day-old WT seedlings on Phytagel and Phytagel C at increasing light concentrations. **B)** Average root penetrance of 6-day-old *sob3-4 esc-8* seedlings on Phytagel and Phytagel C at increasing light concentrations. Standard error is shown for all data sets. In a Welch's t test (unpaired two-tailed t test with unequal variance) compared with Phytagel: $P > 0.05$ = Not Significant (NS), $P \leq 0.01$ = **, and $P \leq 0.001$ = ***.

addition, in all conditions the percentage of root penetrance on 1% Gellan plates was higher than on 1% Phytagel plates (Fig 6 and 6B). The root penetrance data can be explained, at least in part by, the observation that 1% Gellan plates are softer than 1% Phytagel plates (Fig 7). This may also explain the visual difference we see between 1% Phytagel C and 1% Gellan plates (Fig 8).

Since Phytagel is no longer available and Phytagel C is not a viable alternative, a replacement gelling agent needed to be identified. Since 1% Gellan has been shown to reproduce previously published results (Figs 5 and 6), we tested various concentrations to determine a new standard for the lab. 0.6% Gellan plates were able to reproduce the same previously published results (Figs 9 and 10). We replicated the 0.6% experiment with Phytagel C (Fig 11). Even at this lower concentration, Phytagel C was not able to distinguish the *sob3-4 esc-8* phenotype from the WT phenotype.

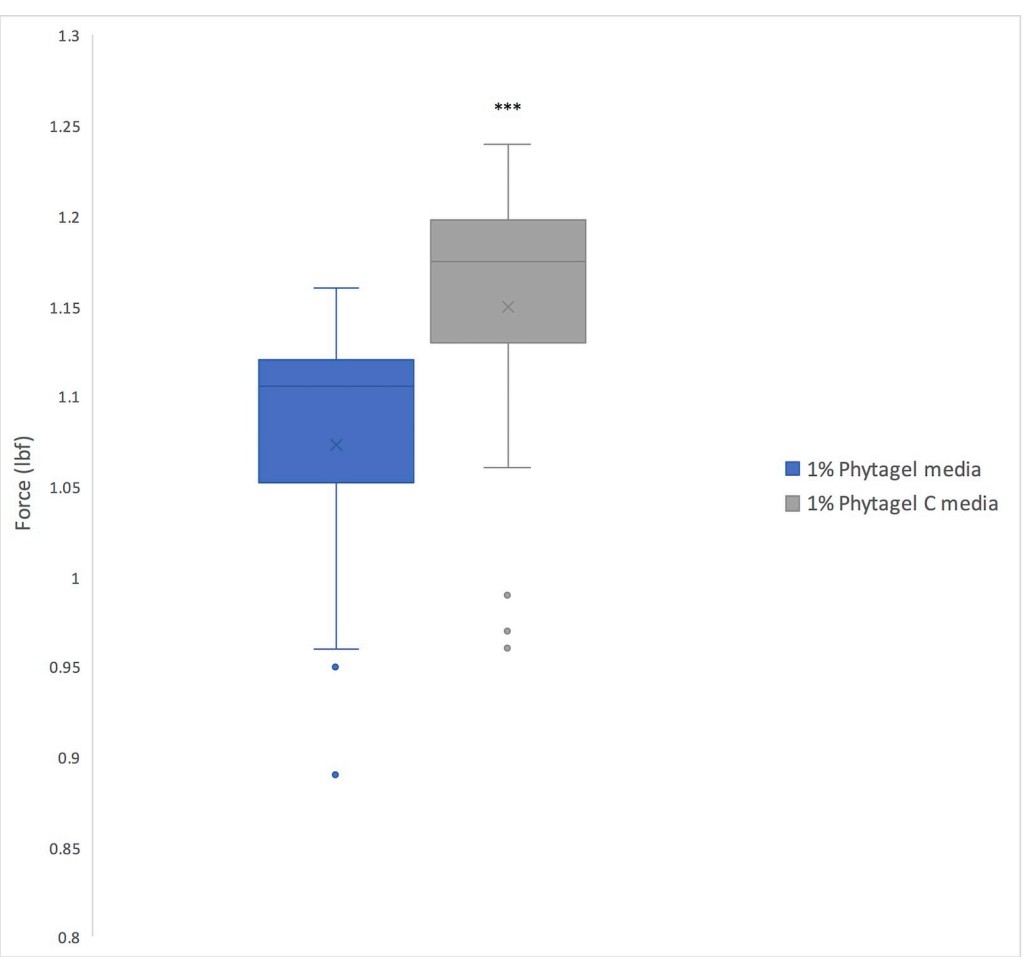

**Fig 4. Force test of 1% Phytagel and 1% Phytagel C plates.** Force required to penetrate a centimeter of 1% growth media containing either Phytagel or Phytagel C as the gelling agent. Three plates for each gelling agent were prepared identically. 10 samples were taken from each plate and averaged. In a Welch's t test (unpaired two-tailed t test with unequal variance) compared with Phytagel: P ≤ 0.0001 = ****.

## Discussion

Seedling phenomics is an important area of research that relies on the reproducibility of growth media plates. This is especially important when seedlings display subtle phenotypes, as in the case of the *A. thaliana* missense allele, *phyB-4*, and the double mutant, *sob3-4 esc-8*. In our study, we aimed to uncover roles that different gelling agents could be playing in *A. thaliana* seedling growth and development. Phytagel produced in America has been used for several studies involving the characterization of subtle seedling phenotypes [3,9,11]. Phytagel produced in China does not replicate these results in at least three distinct ways: fluence-rate-response analyses, root penetrance analyses, and hardness assays.

The disparity in hardness between the 1% Phytagel and 1% Phytagel C plates may explain the difference we see in seedling growth. For example, the hardness of the 1% Phytagel C plates may not be impacting the ability of the seeds to germinate, but it may be impacting the ability of the roots to penetrate the media and allow for proper growth. This may explain why we do not see an impact on germination rates for *sob3-4 esc-8*, but we saw a significant impact on the ability of the *sob3-4 esc-8* roots to penetrate. We suspect that the hardness of the plate may be

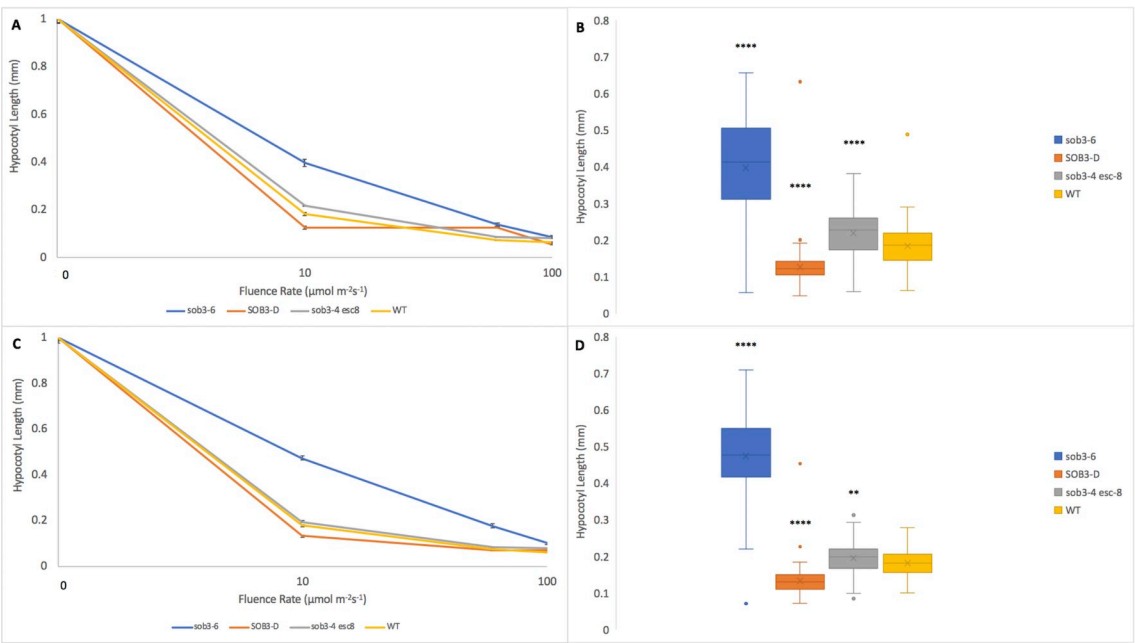

**Fig 5. Fluence rate responses of 6-day-old seedlings on 1% Phytagel and 1% Gellan plates. A)** Fluence rate responses of 6-day-old seedlings that have been normalized to the dark control on plates containing 1% Phytagel media. **B)** Graphical representation of 6-day-old seedlings that have been normalized to the dark control on plates containing 1% Phytagel media at 10 μmol m$^{-2}$s$^{-1}$. **C)** Fluence rate responses of 6-day-old seedlings that have been normalized to the dark control on plates containing 1% Gellan media. **D)** Graphical representation of 6-day-old seedlings that have been normalized to the dark control on plates containing 1% Gellan media at 10 μmol m$^{-2}$s$^{-1}$. Standard error is shown for all data sets. In a Welch's t test (unpaired two-tailed t test with unequal variance) compared with the wild type, P ≤ 0.0001 = ****.

due to a chemical change in the Phytagel C media when exposed to light. The change in seedling growth and development could also be due to a change in water potential within the media. Further testing is needed to determine the cause of the increased hardness of the Phytagel C plates.

Since Phytagel C was not able to give reproducible results, we tested another gelling agent, Gellan, to determine if it was a suitable replacement for Phytagel. We suspected that the cause for similar root penetrance between 1% Gellan and 1% Phytagel plates may be due to likeness in hardness. However, we found that 1% Gellan plates are significantly less hard than 1% Phytagel plates. Chemical and/or osmotic experimentation would possibly clarify the differences in hardness that is seen between the three different gelling agents.

After the aforementioned experiments, the Neff lab decided to replace Phytagel with Gellan for phenotypic experimentation on *A. thaliana* seedlings. This was a necessary replacement, as seedling phenomics were halted in the Neff lab without a reliable gelling agent. Since the 1% protocol had been established with Phytagel, we tested different concentrations of Gellan to establish a new standard. We found that 0.6% Gellan plates gave more uniform visual results than 1%, reproduced previously published results, and is a more cost-effective option. Therefore, the Neff lab has replaced 1% Phytagel with 0.6% Gellan for phenotypic analysis of *v* seedlings.

This study highlights key development differences of *A. thaliana* seedlings on different gelling agents. It doesn't, however, explore the chemical or physical differences that may be causing the changes in growth for plants germinated on Phytagel compared to Phytagel C. Further analyses, such as water potential and metal composition, are needed to determine the cause of

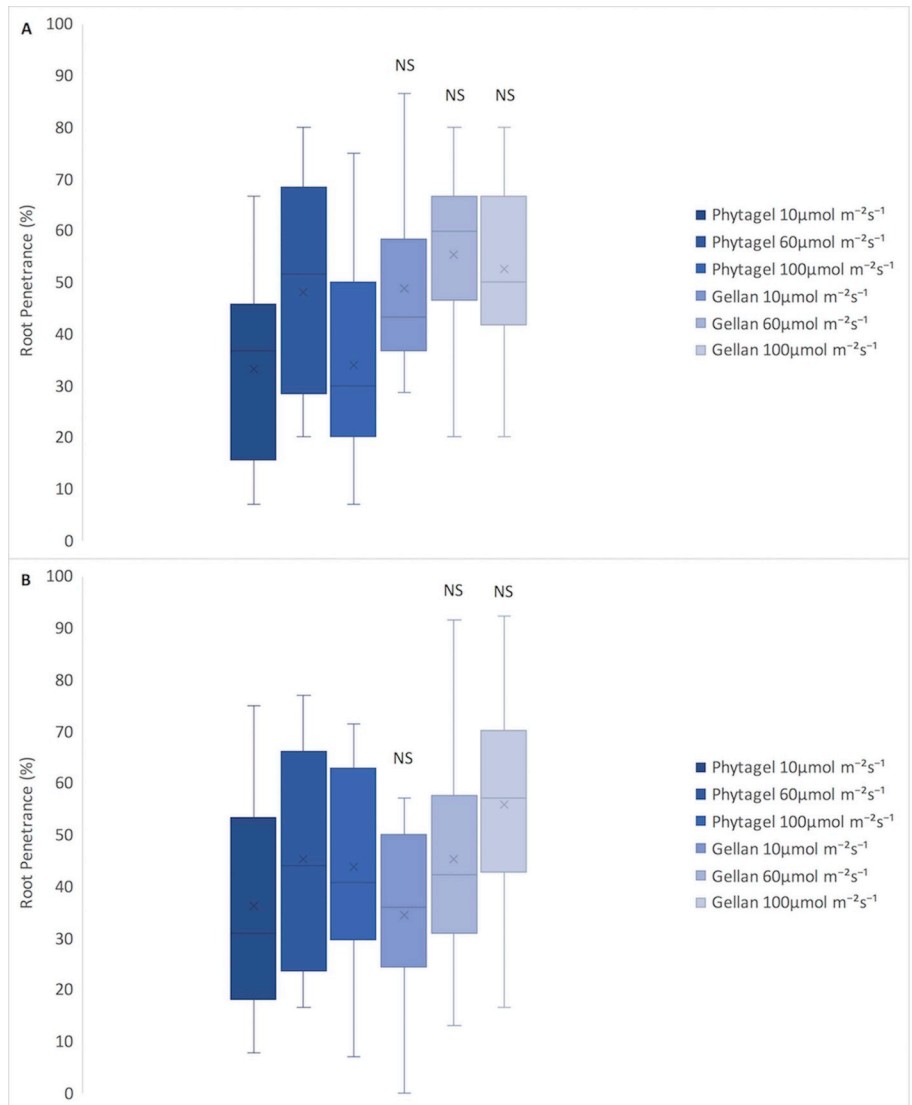

**Fig 6. Average root penetrance of 6-day-old seedlings on 1% Phytagel and 1% Gellan plates. A)** Average root penetrance of 6-day-old WT seedlings on Phytagel and Gellan at increasing light concentrations. **B)** Average root penetrance of 6-day-old *sob3-4 esc-8* seedlings on Phytagel and Gellan at increasing light concentrations. In a Welch's t test (unpaired two-tailed t test with unequal variance) compared with Phytagel: P > 0.05 = Not Significant (NS), P ≤ 0.05 = *.

the changes in Phytagel presented within this manuscript. Additionally, we have not looked at the impact of Phytagel C on tissue culture assays and adult morphologies. Although this information may be of interest to those involved in the production of gelling agents, the main purpose of this study is to report the problems that we have encountered while also providing a viable replacement for seedling morphometric analysis.

## Conclusion

Three gelling agents, Phytagel (no longer commercially available), Phytagel C, and Gellan, were investigated based on their ability to reproduce the subtle phenotype of the *sob3-4 esc-8* double mutant. Fluence-rate-response analysis of 1% Phytagel C plates failed to replicate the

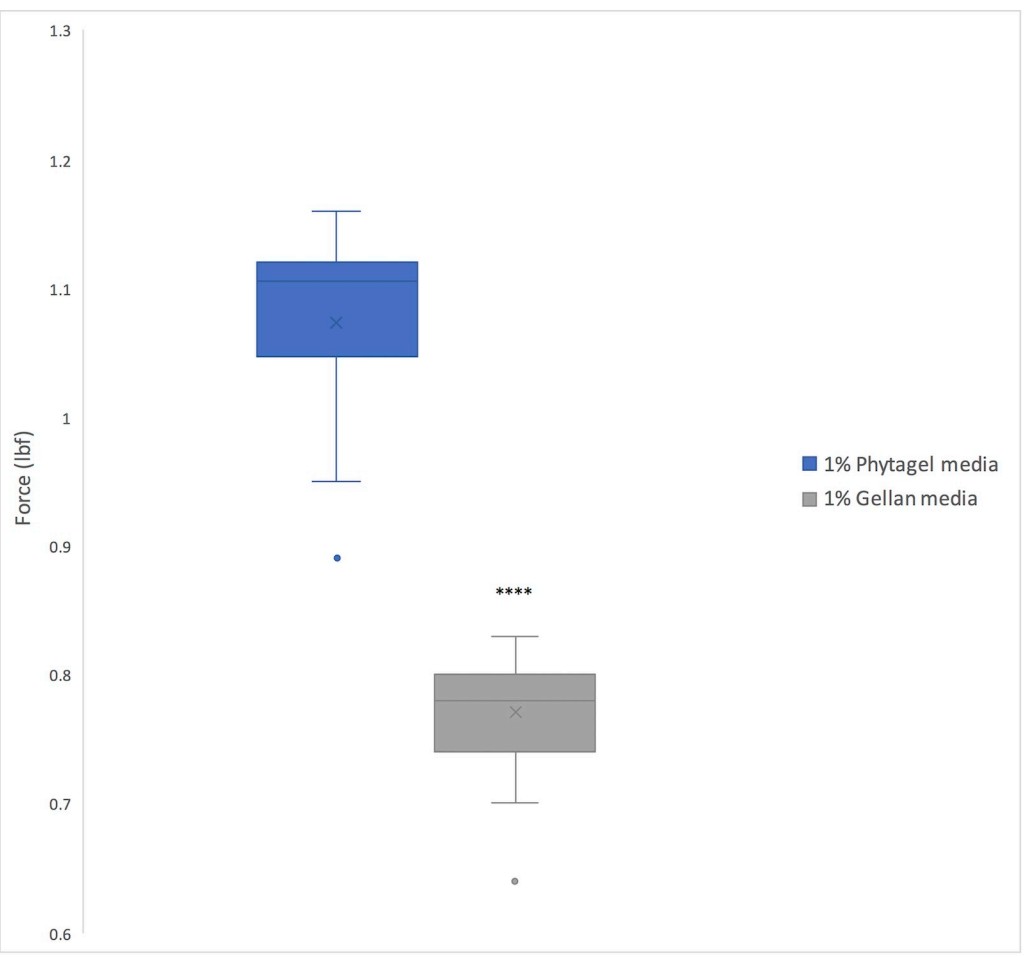

**Fig 7. Force test of 1% Phytagel and 1% Gellan plates.** Force required to penetrate a centimeter of 1% growth media containing either Phytagel or Gellan as the gelling agent. Three plates for each gelling agent were prepared identically. 10 samples were taken from each plate and averaged. In a Welch's t test (unpaired two-tailed t test with unequal variance) compared with Phytagel: P $\leq$ 0.0001 = ****.

*sob3-4 esc-8* subtle seedling phenotype seen on 1% Phytagel. Furthermore, root penetrance analysis showed a significant difference in root penetration between *sob3-4 esc-8* seedlings grown on 1% Phytagel and 1% Phytagel C. Finally, it was found that 1% Phytagel C was significantly harder than 1% Phytagel, which may be causing decreased *sob3-4 esc-8* root penetrance, as well as affecting seedling growth and development.

As a substitute for Phytagel C, 1% Gellan was able to reproduce the subtle phenotype of the *sob3-4 esc-8* double mutant. Furthermore, there was no significant difference in root penetration of the WT or *sob3-4 esc-8* seedlings between 1% Phytagel and 1% Gellan. It was also found that 1% Gellan plates are significantly softer than the 1% Phytagel plates. These observations suggest that Gellan is a suitable replacement for Phytagel. In order to establish a new standard for the lab, we tested different percentages of Gellan media. We found that 0.6% Gellan also reproduces previously published phenotypes and is more cost effective.

The information provided in this manuscript is beneficial to the scientific community as whole, specifically phenomics labs, as it details key problematic differences between gelling agents that should be performing identically (Phytagel and Phytagel C). We also provide labs with additional information on a gelling agent, Gellan, which can replace the use of the no-

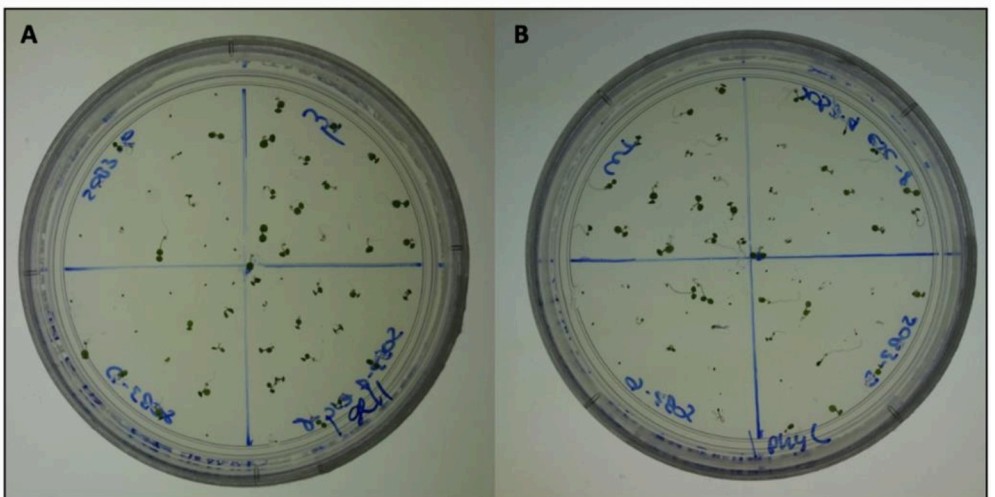

**Fig 8. Photos of seedlings grown on 1% media at 10 μmol m$^{-2}$s$^{-1}$ for six days. A)** 6-day-old seedlings grown on 1% Gellan at 10 μmol m$^{-2}$s$^{-1}$. **B)** 6-day-old seedlings grown on 1% Phytagel C at 10 μmol m$^{-2}$s$^{-1}$.

longer commercially available Phytagel. These data will help to promote consistency of methodologies for better integration of data from different laboratories.

## Methods

### Growth media plates

50mL plates were made with media containing one-half-strength Linsmaier and Skoog modified basal media, 1.5% sucrose (m/v), and the appropriate amount (m/v) of gelling agent. The gelling agents used in this study are: Phytagel (Sigma), Phytagel C (Sigma), and Gellan (Phyto-Technology Laboratories). Experiments were conducted on growth media plates containing 0.6% or 1% of these gelling agents. The lot number for Phytagel is SLBJ1281V. The lot number

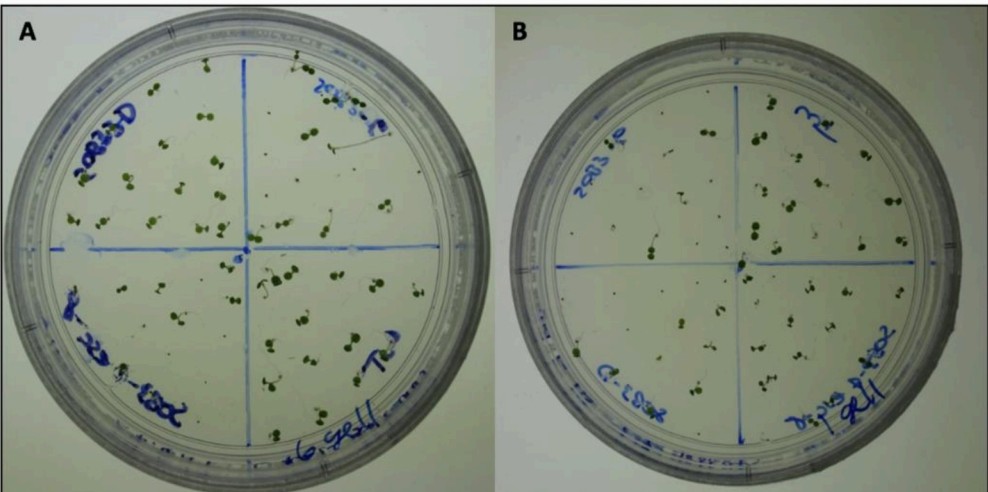

**Fig 9. Photos of 6-day-old seedlings grown on media at 10 μmol m$^{-2}$s$^{-1}$. A)** 6-day-old seedlings grown on 0.6% Gellan at 10 μmol m$^{-2}$s$^{-1}$. **B)** 6-day-old seedlings grown on 1% Gellan at 10 μmol m$^{-2}$s$^{-1}$.

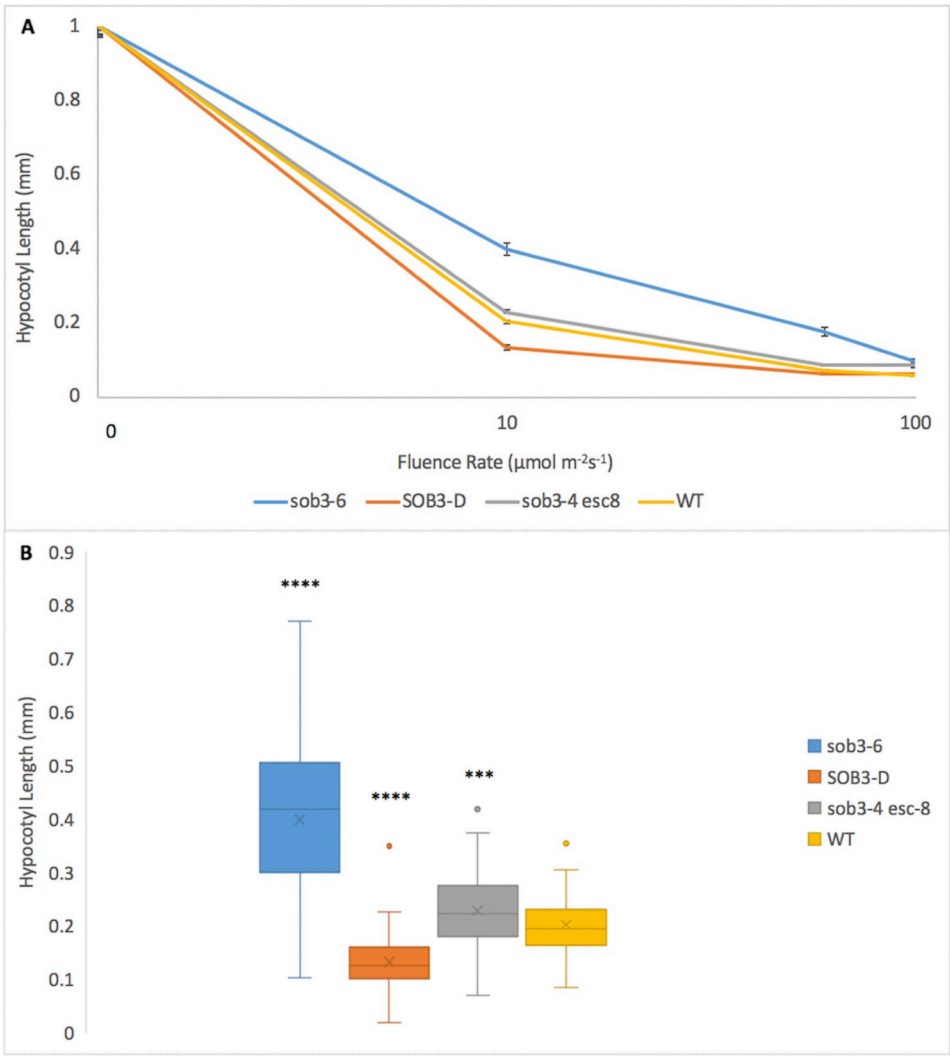

**Fig 10. Fluence rate responses of 6-day-old seedlings plates containing 0.6% Gellan media. A)** Fluence rate responses of 6-day-old seedlings that have been normalized to the dark control on plates containing 0.6% Gellan media. **B)** Graphical representation of 6-day-old seedlings that have been normalized to the dark control on plates containing 0.6% Gellan media at 10 μmol m$^{-2}$s$^{-1}$. Standard error is shown for all data sets. In a Welch's t test (unpaired two-tailed t test with unequal variance) compared with the wild type, $P \leq 0.001 = ^{***}$ and $P \leq 0.0001 = ^{****}$.

for Phytagel produced in China is SLBQ4373V. This was the second batch of product and it was received on October 20, 2016. The first batch was received on July 21, 2016 (lot number SLBQ4319V). The lot number for Gellan is SUS0434045A. Sigma and PhytoTechnology Laboratories do not print the date of production nor the expiration date on their productions.

## Experimental design

15 seeds with known, published phenotypes (WT, *sob3-4 esc-8*, *sob3-6*, and *SOB3*-D) were hand-plated onto evenly divided sectors on each plate. These seeds had been previously sur-face-sterilized, as described below, and were used for up to six weeks post-sterilization. The plates were kept in the dark at 4˚ Celsius for three days to synchronize germination. Post cold and dark treatment, the plates underwent a 12-hour red-light treatment in a growth chamber

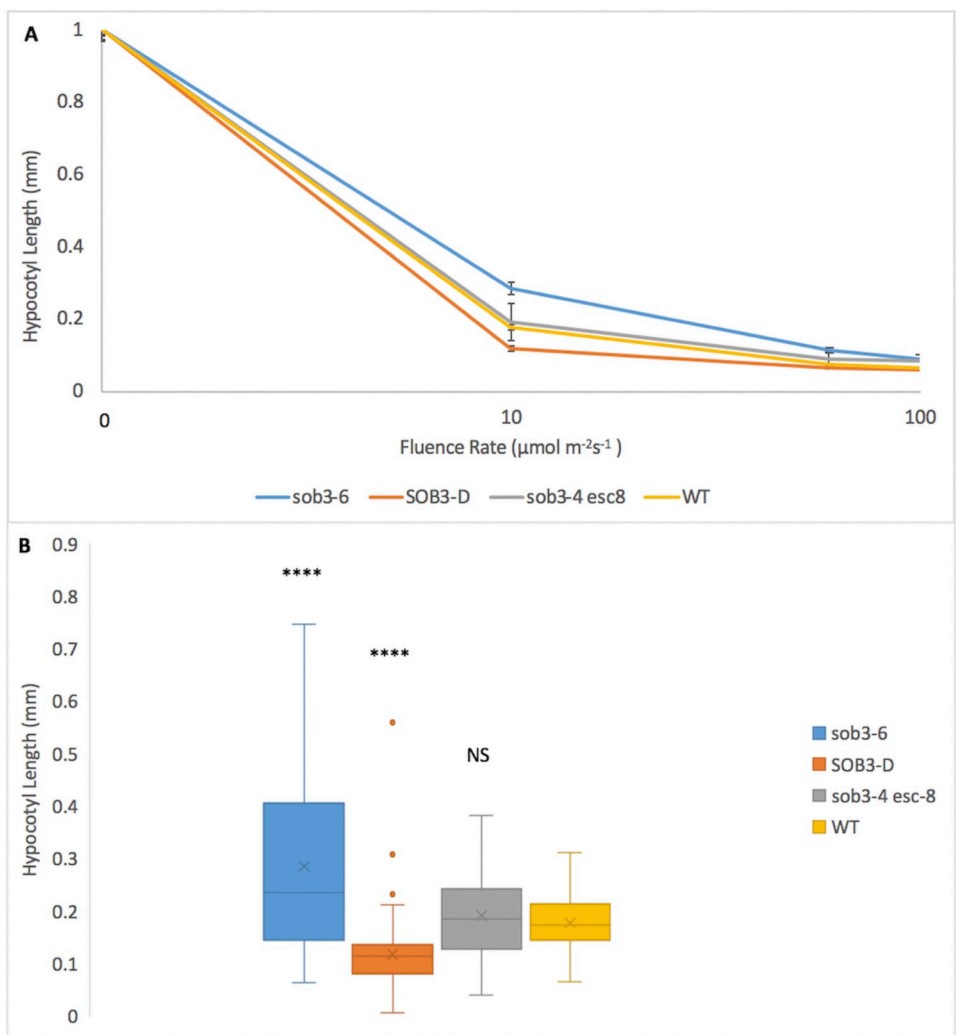

**Fig 11. Fluence rate responses of 6-day-old seedlings on plates containing 0.6% Phytagel C media. A)** Fluence rate responses of 6-day-old seedlings that have been normalized to the dark control on plates containing 0.6% Phytagel C media. **B)** Graphical representation of 6-day-old seedlings that have been normalized to the dark control on plates containing 0.6% Phytagel C media at 10 µmol m$^{-2}$s$^{-1}$. Standard error is shown for all data sets. In a Welch's t test (unpaired two-tailed t test with unequal variance) compared with the wild type, P > 0.05 = Not Significant (NS), P ≤ 0.0001 = ****.

at 25° Celsius. After this red-light treatment, one plate from each gelling agent group was subjected to one of these four light treatments for a total of six days: dark, 10 µmol m$^{-2}$s$^{-1}$, 60 µmol m$^{-2}$s$^{-1}$, and 100 µmol m$^{-2}$s$^{-1}$. At the end of six days, each seedling was analyzed for root penetrance and germination. Root penetrance and germination were recorded separately as binary results: 1 for yes it penetrated/germinated and 0 for no it did not penetrate/germinate. Germination was defined by whether or not there was appearance of the root. For example, a seed that germinated, but did not penetrate the media was recorded as 1/0. After these data were recorded, the seedlings were transferred to transparencies. The transparencies were scanned to the computer and the hypocotyl length of each seedling was measured using ImageJ Software, and is described in detail below. To account for differences in timing of germination, seedling measurements were normalized to the average hypocotyl length of each specific genotype grown in the dark. Eleven replicates were conducted for each light treatment and gelling agent

combination at 1% concentration. Seven replicates were conducted for each light treatment and gelling agent combination at 0.6% concentration.

## Seed sterilization

For sterilization, seeds were placed in microcentrifuge tubes and covered with 75% alcohol containing 0.5% Triton X-100 (v/v) and placed on a shaker for 15 minutes. The liquid was pipetted off and the seeds were covered with 95% alcohol containing 0.5% Triton X-100 (v/v) and shaken for 10 minutes. The liquid was pipetted off and the seeds were covered with 95% alcohol (v/v) and shaken for five minutes. The liquid was pipetting off and the seeds were placed on sterilized filter paper and left to dry in a laminar airflow hood.

## Chamber setup

The light chamber used is E30B (Percival Scientific, Inc.). The different light intensities were achieved through the use of mesh screens. The light intensities were measured before experimentation using a LI-250A mobile spectrophotometer (LI-COR Biosciences). The spectrophotometer was place in the center of the light chamber with the door closed. Periodic measurements were taken to ensure the light intensities were not fluctuating.

## Measuring hypocotyl length via NIH ImageJ software

The transparencies were digitized with a flatbed scanner at 720 dpi. The transparencies included a ruler for measuring a 1mm length to set the parameters for measurements in ImageJ (The NIH). A length of 1mm was established in pixels for each image. The hypocotyls were measured from the top of hypocotyl to the beginning of the roots using the segmented line tool. The same researcher measured all of hypocotyls to ensure no discrepancies would occur in measuring the hypocotyls. The measurements were transferred to an Excel spreadsheet for analysis.

## Hardness testing via FTA probe

A fruit texture analyzer probe (GS-14 Fruit Texture Analyzer, GÜSS Instruments, South Africa) was used to test the force required to penetrate 1cM of media. Ten locations were selected on each plate and tested. Three plates were made at 1% for Phytagel, Phytagel C, and Gellan. The values for like plates were averaged.

## Analysis of data and statistics

Standard error was conducted on all applicable data sets and are included as error bars where appropriate. Welch's t test (unpaired two-tailed t test with unequal variance) was also conducted where appropriate. P values are included as follows: $P > 0.05$ = Not Significant (NS), $P \leq 0.01$ = **, $P \leq 0.001$ = ***, and $P \leq 0.0001$ = ****.

## Supporting information

**S1 Fig. Germination rates on 1% Phytagel and 1% Phytagel C media. A)** Germination rates of WT, *sob3-6*, *SOB3-D*, and *sob3-4 esc-8* at 10 μmol m$^{-2}$s$^{-1}$ on 1% Phytagel and 1% Phytagel C plates. **B)** Germination rates of WT, *sob3-6*, *SOB3-D*, and *sob3-4 esc-8* at 60 μmol m$^{-2}$s$^{-1}$ on 1% Phytagel and Phytagel C plates. **C)** Germination rates of WT, *sob3-6*, *SOB3-D*, and *sob3-4 esc-8* at 100 μmol m$^{-2}$s$^{-1}$ on 1% Phytagel and Phytagel C plates. In a Welch's t test (unpaired two-tailed t test with unequal variance) compared with the wild type, $P > 0.05$ = Not Significant

(NS).
(TIFF)

**S2 Fig. Dark grown seedlings on 1% Phytagel, 1% Phytagel C, and 1% Gellan. A)** Hypocotyl lengths of WT seedlings on different growth media plates. In a Welch's t test (unpaired two-tailed t test with unequal variance) compared Phytagel, $P \leq 0.01 =$ **, $P > 0.05$ Not Significant (NS). **B)** Hypocotyl lengths of *sob3-4 esc-8* seedlings on different growth media plates. In a Welch's t test (unpaired two-tailed t test with unequal variance) compared Phytagel, $P \leq 0.0001 =$ ****, $P > 0.05 =$ Not Significant (NS). **C)** Hypocotyl lengths of *sob3-6* seedlings on different growth media plates. In a Welch's t test (unpaired two-tailed t test with unequal variance) compared Phytagel, $P > 0.05 =$ Not Significant (NS). **D)** Hypocotyl lengths of *SOB3-D* seedlings on different growth media plates. In a Welch's t test (unpaired two-tailed t test with unequal variance) compared Phytagel, $P > 0.05 =$ Not Significant (NS).
(TIFF)

## Acknowledgments

The Smertenko lab at Washington State University provided the original Phytagel needed for this experiment.

Seanna Hewitt of the Dhingra lab at Washington State University provided the Fruit Texture Analyzer (FTA) probe used to test the hardness of the plates.

## Author Contributions

**Conceptualization:** Caitlin N. Jacques.

**Data curation:** Caitlin N. Jacques, Anna K. Hulbert, Shelby Westenskow.

**Formal analysis:** Caitlin N. Jacques, Michael M. Neff.

**Funding acquisition:** Michael M. Neff.

**Investigation:** Caitlin N. Jacques, Anna K. Hulbert, Shelby Westenskow, Michael M. Neff.

**Methodology:** Michael M. Neff.

**Project administration:** Michael M. Neff.

**Supervision:** Michael M. Neff.

**Writing – original draft:** Caitlin N. Jacques.

**Writing – review & editing:** Caitlin N. Jacques, Michael M. Neff.

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
