## [Decision Letter · Decision Letter 0]

10 Mar 2020

PONE-D-20-01374

Production location of the gelling agent Phytagel has a significant impact on Arabidopsis thaliana seedling phenotypic analysis

PLOS ONE

Dear Dr. Neff,

Thank you for submitting your manuscript to PLOS ONE. After careful consideration, we feel that it has merit but does not fully meet PLOS ONE’s publication criteria as it currently stands. Therefore, we invite you to submit a revised version of the manuscript that addresses the points raised during the review process.

We would appreciate receiving your revised manuscript by Apr 24 2020 11:59PM. To enhance the reproducibility of your results, we recommend that if applicable you deposit your laboratory protocols in protocols.io, where a protocol can be assigned its own identifier (DOI) such that it can be cited independently in the future. For instructions see: http://journals.plos.org/plosone/s/submission-guidelines#loc-laboratory-protocols

We look forward to receiving your revised manuscript.

Kind regards,

Robert Hoehndorf, Ph.D.

Academic Editor

PLOS ONE

Additional Editor Comments (if provided):

The manuscript has been reviewed by two experts in the field, and while they find the manuscript and findings of value, they have raised several concerns that should be addressed in a revision.

Journal Requirements:

2)  Our internal editors have looked over your manuscript and determined that it is within the scope of our Plant Phenomics & Precision Agriculture Call for Papers. This collection of papers is headed by a team of Guest Editors for PLOS ONE. The Collection will encompass a diverse range of research articles spanning disciplines, methods and applications.  Additional information can be found on our announcement page: https://plos.io/phenomics.

If you would like your manuscript to be considered for this collection, please let us know in your cover letter and we will ensure that your paper is treated as if you were responding to this call. If you would prefer to remove your manuscript from collection consideration, please specify this in the cover letter.

Reviewers' comments:

Reviewer's Responses to Questions

**Comments to the Author**

1. Is the manuscript technically sound, and do the data support the conclusions?

Reviewer #1: Yes

Reviewer #2: Yes

2. Has the statistical analysis been performed appropriately and rigorously? 

Reviewer #1: Yes

Reviewer #2: Yes

3. Have the authors made all data underlying the findings in their manuscript fully available?

Reviewer #1: Yes

Reviewer #2: Yes

4. Is the manuscript presented in an intelligible fashion and written in standard English?

Reviewer #1: Yes

Reviewer #2: Yes

5. Review Comments to the Author

Reviewer #1: The authors discovered a difference in performance of phenotyping experiments conducted using Phytagel produced in USA vs. the newest Phytagel produced in China. The experiments done were thorough and well presented.

Line 100: Please indicate a more accurate date of change in the production location.

Line 238: Did you test the effects of Phytagel C in reproducing other phenotypes not related to light? Is this difference only upon the application of light treatment?

METHODS:

-Missing the description of the method to measure fluence rate response

Lines 287- 292: Do not call them Agar (or agarose) plates, since you are using agar substitutes, call them growth media plates.

Lines 290-291: Please include year of manufacture and/or expliration date.

Could the effects be due to the use of old gelling agent (Phytagel)?

FIGURES:

Overall, I recommend you use box plots instead of bar plots, so you can identify possible outliers and display more information about your results.

Your results are quite impressive, have you shared them with Sigma company? If so, what do they suggest that could be the cause?

Reviewer #2: This Ms from the Neff lab compares the gelling agent used for phenotypic analyses of Arabidopsis thaliana seedlings. As the company that produces Phytagel changed their provider, the gelling agent changed in its characteristics. As this can strongly influence results and lead to the fact that results are no longer comparable this is a very important finding.

The authors used a very sensitive mutant as a marker to test different gelling agents and tested for several responses, hypocotyl elongation under different light conditions, germination rates and root penetrance.

The authors concluded that mainly the hardness of the media was responsible for the differences observed between the different gelling agents and tested an alternative that can reproduce previous findings.

The paper is well written, the experiments done with care and statistically evaluated.

To make this study more interesting for a larger audience I am missing that other parameters were not checked. Osmotic behavior of the gelling agents and/or different metal ions could have a big impact on the growth of seedlings, especially on the root penetration phenotype. Yet no studies were conducted in this regard - other sensitive mutants could have been included to rule out further problems with the media. Additionally, I am surprised to see that 1,5% sucrose was include in the media, especially for the hypocotyl elongation. Most often for light experiments no sucrose is included in the media. Residual sugars in the media could also have a big impact on the hypocotyl elongation and the overall growth performance. Did the authors see differences in the growth behavior on media without any added sugars?

Hypocotyl length was normalized to the dark control, but I did not find any data if the hypocotyl length in darkness varied under the different media.

As both germination and root penetrance were scored the authors should define more precisely how they scored germination.

In principle this study shows how differences in results between different labs could be explained, and as the phenotypes of the sub3-4 esc-8 double mutant were only detectable on the softer gelling agents phenotypes of other mutants might be more prominent on the harder gelling agent. Therefore the methods and materials section in papers should be precise and papers such as this one are helpful to be aware of changes made by the suppliers.

6. PLOS authors have the option to publish the peer review history of their article (what does this mean?). If published, this will include your full peer review and any attached files.

Reviewer #1: No

Reviewer #2: No

---

## [Author Response · Author response to Decision Letter 0]

24 Apr 2020

Response to Reviewers

This is a resubmission of the manuscript “Production location of the gelling agent Phytagel has a significant impact on Arabidopsis thaliana seedling phenotypic analysis”. We’ve addressed all of the reviewers’ comments to the best of our ability. We hope that you now find this manuscript sufficient for publication in PLoS One. Below you will find a detailed explanation addressing each of the reviewers’ points. 

Reviewer 1:

The authors discovered a difference in performance of phenotyping experiments conducted using Phytagel produced in USA vs. the newest Phytagel produced in China. The experiments done were thorough and well presented.

Authors’ response: Thank you for your comment. We tried to organize the data in the manuscript as efficiently as possible.

Line 100: Please indicate a more accurate date of change in the production location.

Authors’ response: We have reached out to Sigma to determine when the production of Phytagel was moved from the United States to China. The only answer we have received is that manufacturing of Phytagel moved to China in May of 2019. We informed Sigma that we had received two different batches of product produced in China from late 2016. We didn’t receive a response after multiple follow-up attempts. We’ve further clarified when we noticed these changes within the manuscript on lines 110-114. Additionally, the lot numbers for each gelling agent and the dates we received Phytagel C are now on lines 325-329.

Line 238: Did you test the effects of Phytagel C in reproducing other phenotypes not related to light? Is this difference only upon the application of light treatment?

Authors’ response: Thank you for your question. We did not test the effects of Phytagel C on phenotypes that are not related to light, as all of the genotypes that we work with in our lab are involved in light-mediated pathways. This manuscript is intended to be used a guide to other labs by outlining our issues with this gelling agent. Further research is needed to determine the scope of Phytagel C’s effects. This has been further clarified in the discussion on lines 287-289. Additionally, we have included a new supplemental figure (S2) detailing the differences between dark grown seedlings on different 1% media. This information has been added to the results section of the manuscript on lines 139-142.

Methods

Missing description of the method to measure fluence rate response

Authors’ response: Thank you for bringing our attention to this error. How we measured fluence rate response has now been appropriately updated within the manuscript on lines 365-367.

Lines 287-292: Do not call them agar or agarose plates, since you are using agar substitutes, call the growth media plates

Authors’ response: Thank you for your suggestion. We agree with you and have updated the manuscript to reflect this change.

Lines 290-291 Please include year of manufacture and or/expiration date

Authors’ response: The lot numbers have been added to the method section for all gelling agents, as well as the dates we received Phytagel C. Please see lines 325-329.

Could the effects be due to the use of old gelling agent (Phytagel):

Authors’ response: Thank you for your question. The Phytagel C used for our experiments was from a new container of gelling agent received at the end of 2016. We notified Sigma of the issue and they sent us a new container from a different lot number. The issues persisted, which is when we designed our experiments. We have further clarified this within the manuscript to make it clearer to the reader both in the background on lines 110-114 and in the methods section on lines 325-329. 

Figures

Overall, I recommend you use box plots instead of bar plots, so you can identify possible outliers and display more information about your results.

Authors’ response: Thank you for your suggestion. We agree and have updated all bar graphs to box plots. 

Your results are quite impressive, have you shared them with Sigma company? If so, what do they suggest that could be the cause?

Authors’ response: Thank you for your question. Please see the response to the question of old Phytagel above. In summary, we did not receive any suggestion or clarification from Sigma about what could be the cause of our issues. 

Reviewer 2

This Ms from the Neff lab compares the gelling agent used for phenotypic analyses of Arabidopsis thaliana seedlings. As the company that produces Phytagel changed their provider, the gelling agent changed in its characteristics. As this can strongly influence results and lead to the fact that results are no longer comparable this is a very important finding. The authors used a very sensitive mutant as a marker to test different gelling agents and tested for several responses, hypocotyl elongation under different light conditions, germination rates and root penetrance. The authors concluded that mainly the hardness of the media was responsible for the differences observed between the different gelling agents and tested an alternative that can reproduce previous findings. The paper is well written, the experiments done with care and statistically evaluated.

Authors’ response: Thank you for your comments. We took great care in designing and performing the experiments detailed within this manuscript. We are pleased to hear that our data and writing has been well received. We agree that the information provided within this manuscript is important for the scientific community. 

To make this study more interesting for a larger audience I am missing that other parameters were not checked. Osmotic behavior of the gelling agents and/or different metal ions could have a big impact on the growth of seedlings, especially on the root penetration phenotype. Yet no studies were conducted in this regard - other sensitive mutants could have been included to rule out further problems with the media.

Authors’ response: Thank you for your question. The purpose of this manuscript is to outline the problems our lab experienced with Phytagel now produced in China. In doing so, we are notifying the scientific community that their research may be affected by this new formula. This manuscript was not intended to fully describe the reasons behind our findings. We think it is important that this information be received by the community as soon as possible. We have further detailed this explanation in our discussion section on lines 287-289.

Additionally, I am surprised to see that 1,5% sucrose was included in the media, especially for the hypocotyl elongation. Most often for light experiments no sucrose is included in the media. Residual sugars in the media could also have a big impact on the hypocotyl elongation and the overall growth performance. Did the authors see differences in the growth behavior on media without any added sugars? 

Authors’ response: Thank you for your inquiry. We stated within the manuscript that our lab has used a 1% Phytagel media as a standard. However, it wasn’t clear that this standard includes 1.5% sucrose. The mutants used within the manuscript were previously characterized using media that contained 1.5% sucrose. Therefore, we felt it was appropriate to continue to use the same media formula. We have clarified that 1.5% sucrose was part of the previously established 1% Phytagel standard in the background of the manuscript on line 104.

Hypocotyl length was normalized to the dark control, but I did not find any data if the hypocotyl length in darkness varied under the different media.

Authors’ response: Thank you for your comment. We have added a supplemental figure (S2) detailing the findings of the dark grown seedlings on the different media. This has been added to the results section of the manuscript on lines 139-142.

As both germination and root penetrance were scored the authors should define more precisely how they scored germination.

Authors’ response: Thank you for your comment. We have updated this information within the manuscript on line 346.

In principle this study shows how differences in results between different labs could be explained, and as the phenotypes of the sub3-4 esc-8 double mutant was only detectable on the softer gelling agents phenotypes of other mutants might be more prominent on the harder gelling agent. Therefore, the methods and materials section in papers should be precise and papers such as this one are helpful to be aware of changes made by the suppliers.

Authors’ response: Thank you for your comment. We agree, and we hope our changes to the manuscript allow for a more precise explanation of our methodology and findings.

---

## [Editor Report · Decision Letter 1]

29 Apr 2020

Production location of the gelling agent Phytagel has a significant impact on Arabidopsis thaliana seedling phenotypic analysis

PONE-D-20-01374R1

Dear Dr. Neff,

We are pleased to inform you that your manuscript has been judged scientifically suitable for publication and will be formally accepted for publication once it complies with all outstanding technical requirements.

With kind regards,

Robert Hoehndorf, Ph.D.

Academic Editor

PLOS ONE
---

## [Editor Report · Acceptance letter]

4 May 2020

PONE-D-20-01374R1 

Production location of the gelling agent Phytagel has a significant impact on *Arabidopsis thaliana* seedling phenotypic analysis  

Dear Dr. Neff:

I am pleased to inform you that your manuscript has been deemed suitable for publication in PLOS ONE. Congratulations! Your manuscript is now with our production department. 

With kind regards,

on behalf of

Dr. Robert Hoehndorf 

Academic Editor

PLOS ONE